# Debiased Contrastive Learning

**Ching-Yao Chuang, Joshua Robinson, Lin Yen-Chen**
**Antonio Torralba, Stefanie Jegelka**
CSAIL, Massachusetts Institute of Technology
Cambridge, MA 02139, USA
`{cychuang, joshrob, yenchenl, torralba, stefje}@mit.edu`

## Abstract

A prominent technique for self-supervised representation learning has been to contrast semantically similar and dissimilar pairs of samples. Without access to labels, dissimilar (negative) points are typically taken to be randomly sampled datapoints, implicitly accepting that these points may, in reality, actually have the same label. Perhaps unsurprisingly, we observe that sampling negative examples from truly different labels improves performance, in a synthetic setting where labels are available. Motivated by this observation, we develop a debiased contrastive objective that corrects for the sampling of same-label datapoints, even without knowledge of the true labels. Empirically, the proposed objective consistently outperforms the state-of-the-art for representation learning in vision, language, and reinforcement learning benchmarks. Theoretically, we establish generalization bounds for the downstream classification task.

## 1   Introduction

Learning good representations without supervision has been a long-standing goal of machine learning. One such approach is *self-supervised learning*, where auxiliary learning objectives leverage labels that can be observed without a human labeler. For instance, in computer vision, representations can be learned from colorization [45], predicting transformations [10, 32], or generative modeling [24, 14, 3]. Remarkable success has also been achieved in the language domain [30, 25, 8].

Recently, self-supervised representation learning algorithms that use a contrastive loss have out-performed even supervised learning [15, 28, 19, 18, 2]. The key idea of *contrastive learning* is to contrast semantically similar (positive) and dissimilar (negative) pairs of data points, encouraging the representations $f$ of similar pairs $(x, x^+)$ to be close, and those of dissimilar pairs $(x, x^-)$ to be more orthogonal [33, 2]:

$$\mathbb{E}_{x, x^+, \{x_i^-\}_{i=1}^N} \left[ -\log \frac{e^{f(x)^T f(x^+)}}{e^{f(x)^T f(x^+)} + \sum_{i=1}^N e^{f(x)^T f(x_i^-)}} \right]. \tag{1}$$

In practice, the expectation is replaced by the empirical estimate. For each training data point $x$, it is common to use one positive example, e.g., derived from perturbations, and $N$ negative examples $x_i^-$. Since true labels or true semantic similarity are typically not available, negative counterparts $x_i^-$ are commonly drawn uniformly from the training data. But, this means it is possible that $x^-$ is actually similar to $x$, as illustrated in Figure 1. This phenomenon, which we refer to as *sampling bias*, can empirically lead to significant performance drop. Figure 2 compares the accuracy for learning with this bias, and for drawing $x_i^-$ from data with truly different labels than $x$; we refer to this method as *unbiased* (further details in Section 5.1).

However, the ideal unbiased objective is unachievable in practice since it requires knowing the labels, i.e., *supervised* learning. This dilemma poses the question whether it is possible to reduce the gap

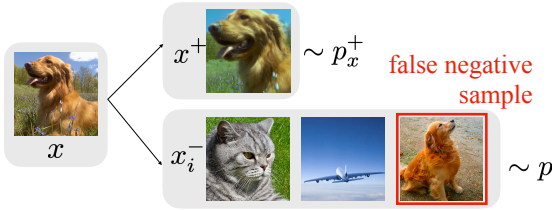

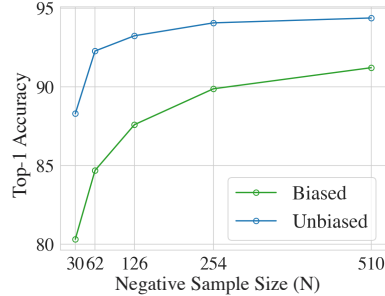

Figure 1: **"Sampling bias":** The common practice of drawing negative examples $x_i^-$ from the data distribution $p(x)$ may result in $x_i^-$ that are actually similar to $x$.

Figure 2: **Sampling bias leads to performance drop:** Results on CIFAR-10 for drawing $x_i^-$ from $p(x)$ (biased) and from data with different labels, i.e., truly semantically different data (unbiased).

between the ideal objective and standard contrastive learning, without supervision. In this work, we demonstrate that this is indeed possible, while still assuming only access to unlabeled training data and positive examples. In particular, we develop a correction for the sampling bias that yields a new, modified loss that we call *debiased contrastive loss*. The key idea underlying our approach is to indirectly approximate the distribution of negative examples. The new objective is easily compatible with any algorithm that optimizes the standard contrastive loss. Empirically, our approach improves over the state of the art in vision, language and reinforcement learning benchmarks.

Our theoretical analysis relates the debiased contrastive loss to supervised learning: optimizing the debiased contrastive loss corresponds to minimizing an upper bound on a supervised loss. This leads to a generalization bound for the supervised task, when training with the debiased contrastive loss.

In short, this work makes the following contributions:

- We develop a new, *debiased contrastive objective* that corrects for the sampling bias of negative examples, while only assuming access to positive examples and the unlabeled data;
- We evaluate our approach via experiments in vision, language, and reinforcement learning;
- We provide a theoretical analysis of the debiased contrastive representation with generalization guarantees for a resulting classifier.

## 2   Related Work

**Contrastive Representation Learning.**   The contrastive loss has recently become a prominent tool in unsupervised representation learning, leading to state-of-the-art results. The main difference between different approaches to contrastive learning lies in their strategy for obtaining positive pairs. Examples in computer vision include random cropping and flipping [33], or different views of the same scene [40]. Chen et al. [2] extensively study verious data augmentation methods. For language, Logeswaran and Lee [28] treat the context sentences as positive samples to efficiently learn sentence representations. Srinivas et al. [37] improve the sample efficiency of reinforcement learning with representations learned via the contrastive loss. Computational efficiency has been improved by maintaining a dictionary of negative examples [18, 4]. Concurrently, Wang and Isola [42] analyze the asymptotic contrastive loss and propose new metrics to measure the representation quality. All of these works sample negative examples from $p(x)$.

Arora et al. [1] theoretically analyze the effect of contrastive representation learning on a downstream, "average" classification task and provide a generalization bound for the standard objective. They too point out the sampling bias as a problem, but do not propose any models to address it.

**Positive-unlabeled Learning.**   Since we approximate the contrastive loss with only unlabeed data from $p(x)$ and positive examples, our work is also related to *Positive-Unlabeled* (PU) learning, i.e., learning from only positive (P) and unlabeled (U) data. Common applications of PU learning are retrieval or outlier detection [13, 12, 11]. Our approach is related to *unbiased PU learning*, where the unlabeled data is used as negative examples, but down-weighted appropriately [26, 12, 11].

While these works focus on zero-one losses, we here address the contrastive loss, where existing PU estimators are not directly applicable.

## 3   Setup and Sampling Bias in Contrastive Learning

Contrastive learning assumes access to semantically similar pairs of data points $(x, x^+)$, where $x$ is drawn from a data distribution $p(x)$ over $\mathcal{X}$. The goal is to learn an embedding $f : \mathcal{X} \to \mathbb{R}^d$ that maps an observation $x$ to a point on a hypersphere with radius $1/t$, where $t$ is the temperature scaling hyperparameter. Without loss of generality, we set $t = 1$ for all theoretical results.

Similar to [1], we assume an underlying set of discrete latent classes $\mathcal{C}$ that represent semantic content, i.e., similar pairs $(x, x^+)$ have the same latent class. Denoting the distribution over classes by $\rho(c)$, we obtain the joint distribution $p_{x,c}(x, c) = p(x|c)\rho(c)$. Let $h : \mathcal{X} \to \mathcal{C}$ be the function assigning the latent class labels. Then $p_x^+(x') = p(x'|h(x') = h(x))$ is the probability of observing $x'$ as a positive example for $x$ and $p_x^-(x') = p(x'|h(x') \neq h(x))$ the probability of a negative example. We assume that the class probabilities $\rho(c) = \tau^+$ are uniform, and let $\tau^- = 1 - \tau^+$ be the probability of observing any different class.

Note that to remain unsupervised in practice, our method and other contrastive losses only sample from the data distribution and a "surrogate" positive distribution, mimicked by data augmentations or context sentences [2, 28].

### 3.1   Sampling Bias

Intuitively, the contrastive loss will provide most informative representations for downstream classification tasks if the positive and negative pairs correspond to the desired latent classes. Hence, the ideal loss to optimize would be

$$L_{\text{Unbiased}}^N(f) = \mathbb{E}_{\substack{x \sim p, x^+ \sim p_x^+ \\ x_i^- \sim p_x^-}} \left[ - \log \frac{e^{f(x)^T f(x^+)}}{e^{f(x)^T f(x^+)} + \frac{Q}{N} \sum_{i=1}^N e^{f(x)^T f(x_i^-)}} \right], \qquad (2)$$

which we will refer to as the *unbiased loss*. Here, we introduce a weighting parameter $Q$ for the analysis. When the number $N$ of negative examples is finite, we set $Q = N$, in agreement with the standard contrastive loss. In practice, however, $p_x^-(x_i^-) = p(x_i^-|h(x_i^-) \neq h(x))$ is not accessible. The standard approach is thus to sample negative examples $x_i^-$ from the (unlabeled) $p(x)$ instead. We refer to the resulting loss as the *biased* loss $L_{\text{Biased}}^N$. When drawn from $p(x)$, the sample $x_i^-$ will come from the same class as $x$ with probability $\tau^+$.

Lemma 1 shows that in the limit, the standard loss $L_{\text{Biased}}^N$ upper bounds the ideal, unbiased loss.

**Lemma 1.** *For any embedding $f$ and finite $N$, we have*

$$L_{\text{Biased}}^N(f) \geq L_{\text{Unbiased}}^N(f) + \mathbb{E}_{x \sim p} \left[ 0 \wedge \log \frac{\mathbb{E}_{x^+ \sim p_x^+} \exp f(x)^\top f(x^+)}{\mathbb{E}_{x^- \sim p_x^-} \exp f(x)^\top f(x^-)} \right] - e^{3/2} \sqrt{\frac{\pi}{2N}}. \qquad (3)$$

*where $a \wedge b$ denotes the minimum of two real numbers $a$ and $b$.*

Recent works often use large $N$, e.g., $N = 65536$ in [18], making the last term negligible. While, in general, minimizing an upper bound on a target objective is a reasonable idea, two issues arise here: (1) the smaller the unbiased loss, the larger is the second term, widening the gap; and (2) the empirical results in Figure 2 and Section 5 show that minimizing the upper bound $L_{\text{Biased}}^N$ and minimizing the ideal loss $L_{\text{Unbiased}}^N$ can result in very different learned representations.

## 4   Debiased Contrastive Loss

Next, we derive a loss that is closer to the ideal $L_{\text{Unbiased}}^N$, while only having access to positive samples and samples from $p$. Figure 2 shows that the resulting embeddings are closer to those learned with $L_{\text{Unbiased}}^N$. We begin by decomposing the data distribution as

$$p(x') = \tau^+ p_x^+(x') + \tau^- p_x^-(x').$$

An immediate approach would be to replace $p_x^-$ in $L_{\text{Unbiased}}^N$ with $p_x^-(x') = (p(x') - \tau^+ p_x^+(x'))/\tau^-$ and then use the empirical counterparts for $p$ and $p_x^+$. The resulting objective can be estimated with samples from only $p$ and $p_x^+$, but is computationally expensive for large $N$:

$$\frac{1}{(\tau^-)^N} \sum_{k=0}^N \binom{N}{k} (-\tau^+)^k \mathop{\mathbb{E}}_{\substack{x \sim p, x^+ \sim p_x^+ \\ \{x_i^-\}_{i=1}^k \sim p_x^+ \\ \{x_i^-\}_{i=k+1}^N \sim p}} \left[ -\log \frac{e^{f(x)^T f(x^+)}}{e^{f(x)^T f(x^+)} + \sum_{i=1}^N e^{f(x)^T f(x_i^-)}} \right], \quad (4)$$

where $\{x_i^-\}_{i=k}^j = \emptyset$ if $k > j$. It also demands at least $N$ positive samples. To obtain a more practical form, we consider the asymptotic form as the number $N$ of negative examples goes to infinity.

**Lemma 2.** *For fixed $Q$ and $N \to \infty$, it holds that*

$$\mathop{\mathbb{E}}_{\substack{x \sim p, x^+ \sim p_x^+ \\ \{x_i^-\}_{i=1}^N \sim p_x^-}} \left[ -\log \frac{e^{f(x)^T f(x^+)}}{e^{f(x)^T f(x^+)} + \frac{Q}{N} \sum_{i=1}^N e^{f(x)^T f(x_i^-)}} \right] \quad (5)$$

$$\longrightarrow \mathop{\mathbb{E}}_{\substack{x \sim p \\ x^+ \sim p_x^+}} \left[ -\log \frac{e^{f(x)^T f(x^+)}}{e^{f(x)^T f(x^+)} + \frac{Q}{\tau^-} \left( \mathbb{E}_{x^- \sim p}[e^{f(x)^T f(x^-)}] - \tau^+ \mathbb{E}_{v \sim p_x^+}[e^{f(x)^T f(v)}] \right)} \right]. \quad (6)$$

The limiting objective (6), which we denote by $\widetilde{L}_{\text{Debiased}}^Q$, still samples examples $x^-$ from $p$, but corrects for that with additional positive samples $v$. This essentially reweights positive and negative terms in the denominator.

The empirical estimate of $\widetilde{L}_{\text{Debiased}}^Q$ is much easier to compute than the straightforward objective (5). With $N$ samples $\{u_i\}_{i=1}^N$ from $p$ and $M$ samples $\{v_i\}_{i=1}^M$ from $p_x^+$, we estimate the expectation of the second term in the denominator as

$$g(x, \{u_i\}_{i=1}^N, \{v_i\}_{i=1}^M) = \max \left\{ \frac{1}{\tau^-} \left( \frac{1}{N} \sum_{i=1}^N e^{f(x)^T f(u_i)} - \tau^+ \frac{1}{M} \sum_{i=1}^M e^{f(x)^T f(v_i)} \right), \; e^{-1/t} \right\}. \quad (7)$$

We constrain the estimator $g$ to be greater than its theoretical minimum $e^{-1/t} \le \mathbb{E}_{x^- \sim p_x^-} e^{f(x)^T f(x_i^-)}$ to prevent calculating the logarithm of a negative number. The resulting population loss with fixed $N$ and $M$ per data point is

$$L_{\text{Debiased}}^{N,M}(f) = \mathop{\mathbb{E}}_{\substack{x \sim p; \, x^+ \sim p_x^+ \\ \{u_i\}_{i=1}^N \sim p^N \\ \{v_i\}_{i=1}^N \sim p_x^{+M}}} \left[ -\log \frac{e^{f(x)^T f(x^+)}}{e^{f(x)^T f(x^+)} + N g\left( x, \{u_i\}_{i=1}^N, \{v_i\}_{i=1}^M \right)} \right], \quad (8)$$

where, for simplicity, we set $Q$ to the finite $N$. The class prior $\tau^+$ can be estimated from data [21, 6] or treated as a hyperparameter. Theorem 3 bounds the error due to finite $N$ and $M$ as decreasing with rate $\mathcal{O}(N^{-1/2} + M^{-1/2})$.

**Theorem 3.** *For any embedding $f$ and finite $N$ and $M$, we have*

$$\left| \widetilde{L}_{\text{Debiased}}^N(f) - L_{\text{Debiased}}^{N,M}(f) \right| \le \frac{e^{3/2}}{\tau^-} \sqrt{\frac{\pi}{2N}} + \frac{e^{3/2} \tau^+}{\tau^-} \sqrt{\frac{\pi}{2M}}. \quad (9)$$

Empirically, the experiments in Section 5 also show that larger $N$ and $M$ consistently lead to better performance. In the implementations, we use a full empirical estimate for $L_{\text{Debiased}}^{N,M}$ that averages the loss over $T$ points $x$, for finite $N$ and $M$.

## 5 Experiments

In this section, we evaluate our new objective $L_{\text{Debiased}}^N$ empirically, and compare it to the standard loss $L_{\text{Biased}}^N$ and the ideal loss $L_{\text{Unbiased}}^N$. In summary, we observe the following: (1) the new loss outperforms state of the art contrastive learning on vision, language and reinforcement learning benchmarks; (2) the learned embeddings are closer to those of the ideal, unbiased objective; (3) both larger $N$ and large $M$ improve the performance; even one more positive example than the standard $M = 1$ can help noticeably. Detailed experimental settings can be found in the appendix. The code is available at https://github.com/chingyaoc/DCL.

```
1  # pos: exponential for positive example
2  # neg: sum of exponentials for negative examples
3  # N  : number of negative examples
4  # t  : temperature scaling
5  # tau_plus: class probability
6
7  standard_loss = -log(pos / (pos + neg))
8  Ng = max((-N * tau_plus * pos + neg) / (1-tau_plus), N * e**(-1/t))
9  debiased_loss = -log(pos / (pos + Ng))
```

Figure 3: **Pseudocode for debiased objective with** $M = 1$**.** The implementation only requires a small modification of the code. We can simply extend the code to debiased objective with $M > 1$ by changing the pos in line 8 with an average of exponentials for $M$ positive samples.

## 5.1 CIFAR10 and STL10

First, for CIFAR10 [27] and STL10 [7], we implement SimCLR [2] with ResNet-50 [17] as the encoder architecture and use the Adam optimizer [23] with learning rate $0.001$. Following [2], we set the temperature to $t = 0.5$ and the dimension of the latent vector to $128$. All the models are trained for $400$ epochs and evaluated by training a linear classifier after fixing the learned embedding. Detailed experimental settings can be found in Appendix B.

To understand the effect of the sampling bias, we additionally consider an estimate of the ideal $L_{\text{Unbiased}}^N$, which is a *supervised* version of the standard loss, where negative examples $x_i^-$ are drawn from the true $p_x^-$, i.e., using known classes. Since STL10 is not fully labeled, we only use the unbiased objective on CIFAR10.

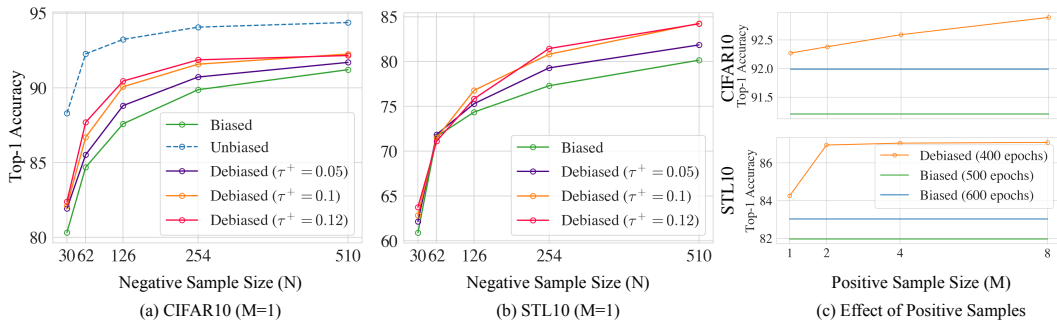

Figure 4: **Classification accuracy on CIFAR10 and STL10.** (a,b) Biased and Debiased ($M = 1$) SimCLR with different negative sample size $N$ where $N = 2(BatchSize - 1)$. (c) Comparison with biased SimCLR with 50% more training epochs (600 epochs) while fixing the training epoch for Debiased ($M \geq 1$) SimCLR to 400 epochs.

**Debiased Objective with** $M = 1$**.** For a fair comparison, i.e., no possible advantage from additional samples, we first examine our debiased objective with positive sample size $M = 1$ by setting $v_1 = x^+$. Then, our approach uses exactly the same data batch as the biased baseline. The debiased objective can be easily implemented by a slight modification of the code as Figure 3 shows. The results with different $\tau^+$ are shown in Figure 4(a,b). Increasing $\tau^+$ in Objective (7) leads to more correction, and gradually improves the performance in both benchmarks for different $N$. Remarkably, with only a slight modification to the loss, we improve the accuracy of SimCLR on STL10 by $4.26\%$. The performance of the debiased objective also improves by increasing the negative sample size $N$.

**Debiased Objective with** $M \geq 1$**.** By Theorem 3, a larger positive sample size $M$ leads to a better estimate of the loss. To probe its effect, we increase $M$ for all $x$ (e.g., $M$ times data augmentation) while fixing $N = 256$ and $\tau^+ = 0.1$. Since increasing $M$ requires additional computation, we compare our debiased objective with biased SimCLR trained for 50% more epochs (600 epochs). The results for $M = 1, 2, 4, 8$ are shown in Figure 4(c), and indicate that the performance of the debiased objective can indeed be further improved by increasing the number of positive samples. Surprisingly,

with only one additional positive sample, the top-1 accuracy on STL10 can be significantly improved. We can also see that the debiased objective ($M > 1$) even outperforms a biased baseline trained with 50% more epochs.

Figure 5 shows t-SNE visualizations of the representations learned by the biased and debiased objectives ($N = 256$) on CIFAR10. The debiased contrastive loss leads to better class separation than the contrastive loss, and the result is closer to that of the ideal, unbiased loss.

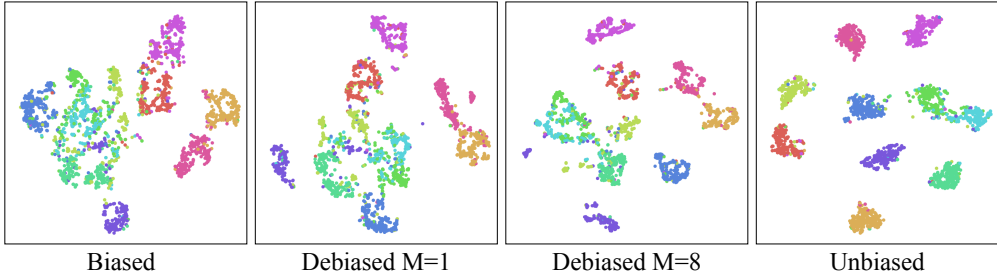

| Biased | Debiased M=1 | Debiased M=8 | Unbiased |

Figure 5: **t-SNE visualization of learned representations on CIFAR10.** Classes are indicated by colors. The debiased objective ($\tau^+ = 0.1$) leads to better data clustering than the (standard) biased loss; its effect is closer to the supervised unbiased objective.

## 5.2 ImageNet-100

Following [40], we test our approach on ImageNet-100, a randomly chosen subset of 100 classes of Imagenet. Compared to CIFAR10, ImageNet-100 has more classes and hence smaller class probabilities $\tau^+$. We use contrastive multiview coding (CMC) [40] as our contrastive learning baseline, and $M = 1$ for a fair comparison. The results in Table 1 show that, although $\tau^+$ is small, our debiased objective still improves over the biased baseline.

| Objective | Top-1 | Top-5 |
|---|---|---|
| Biased (CMC) | 73.58 | 92.06 |
| Debiased ($\tau^+ = 0.005$) | 73.86 | 91.86 |
| Debiased ($\tau^+ = 0.01$) | **74.6** | **92.08** |

Table 1: **ImageNet-100** Top-1 and Top-5 classification results.

## 5.3 Sentence Embeddings

Next, we test the debiased objective for learning sentence embeddings. We use the BookCorpus dataset [25] and examine six classification tasks: movie review sentiment (MR) [35], product reviews (CR) [20], subjectivity classification (SUBJ) [34], opinion polarity (MPQA) [43], question type classification (TREC) [41], and paraphrase identification (MSRP) [9]. Our experimental settings follow those for quick-thought (QT) vectors in [28]. In contrast to vision tasks, positive pairs here are chosen as neighboring sentences, which can form a different positive distribution from data augmentation. The minibatch of QT is constructed with a contiguous set of sentences, hence we can use the preceding and succeeding sentences as positive samples ($M = 2$). We retrain each model 3 times and show the average in Table 2. The debiased objective improves over the baseline in 4 out of 6 downstream tasks, verifying that our objective also works for a different modality.

| Objective | MR | CR | SUBJ | MPQA | TREC | MSRP | |
|---|---|---|---|---|---|---|---|
| | | | | | | (Acc) | (F1) |
| Biased (QT) | **76.8** | 81.3 | 86.6 | 93.4 | **89.8** | 73.6 | 81.8 |
| Debiased ($\tau^+ = 0.005$) | 76.5 | 81.5 | 86.6 | 93.6 | 89.1 | 74.2 | 82.3 |
| Debiased ($\tau^+ = 0.01$) | 76.2 | **82.9** | **86.9** | **93.7** | 89.1 | **74.7** | **82.7** |

Table 2: **Classification accuracy on downstream tasks.** We compare sentence representations on six classification tasks. 10-fold cross validation is used in testing the performance for binary classification tasks (MR, CR, SUBJ, MPQA).

## 5.4 Reinforcement Learning

Lastly, we consider reinforcement learning. We follow the experimental settings of Contrastive Unsupervised Representations for Reinforcement Learning (CURL) [37] to perform image-based policy control on top of the learned contrastive representations. Similar to vision tasks, the positive pairs are two different augmentations of the same image. We again set $M = 1$ for a fair comparison. Methods are tested at 100k environment steps on the DeepMind control suite [39], which consists of several continuous control tasks. We retrain each model 3 times and show the mean and standard deviation in Table 3. Our method consistently outperforms the state-of-the-art baseline (CURL) in different control tasks, indicating that correcting the sampling bias also improves the performance and data efficiency of reinforcement learning. In several tasks, the debiased approach also has smaller variance. With more positive examples ($M = 2$), we obtain further improvements.

| Objective | Finger Spin | Cartpole Swingup | Reacher Easy | Cheetah Run | Walker Walk | Ball in Cup Catch |
|---|---|---|---|---|---|---|
| Biased (CURL) | 310±33 | 850±20 | 918±96 | 266±41 | 623±120 | 928±47 |
| *Debiased Objective with $M = 1$* | | | | | | |
| Debiased ($\tau^+ = 0.01$) | 324±34 | 843±30 | **927±99** | **310±12** | **626±82** | 937±9 |
| Debiased ($\tau^+ = 0.05$) | 308±57 | **866±7** | 916±114 | 284±20 | 613±22 | 945±13 |
| Debiased ($\tau^+ = 0.1$) | **364±36** | 860±4 | 868±177 | 302±29 | 594±33 | **951±11** |
| *Debiased Objective with $M = 2$* | | | | | | |
| Debiased ($\tau^+ = 0.01$) | 330±10 | 858±10 | 754±179 | 286±20 | **746±93** | 949±5 |
| Debiased ($\tau^+ = 0.1$) | **381±24** | 864±6 | 904±117 | 303±5 | 671±75 | **957±5** |

Table 3: **Scores achieved by biased and debiased objectives.** Our debiased objective outperforms the biased baseline (CURL) in all the environments, and often has smaller variance.

## 5.5 Discussion

**Class Distribution:** Our theoretical results assume that the class distribution $\rho$ is close to uniform. In reality, this is often not the case, e.g., in our experiments, CIFAR10 and Imagenet-100 are the only two datasets with perfectly balanced class distributions. Nevertheless, our debiased objective still improves over the baselines even when the classes are not well balanced, indicating that the objective is robust to violations of the class balance assumption.

**Positive Distribution:** Even if we approximate the true positive distribution with a surrogate positive distribution, our debiased objective still consistently improves over the baselines. It is an interesting avenue of future work to adopt our debiased objective to a semi-supervised learning setting [44] where true positive samples are accessible.

## 6 Theoretical Analysis: Generalization Implications for Classification Tasks

Next, we relate the debiased contrastive objective to a supervised loss, and show how our contrastive learning approach leads to a generalization bound for a downstream supervised learning task.

We consider a supervised classification task $\mathcal{T}$ with $K$ classes $\{c_1, \ldots, c_K\} \subseteq \mathcal{C}$. After contrastive representation learning, we fix the representations $f(x)$ and then train a linear classifier $q(x) = Wf(x)$ on task $\mathcal{T}$ with the standard multiclass softmax cross entropy loss $L_{\text{Softmax}}(\mathcal{T}, q)$. Hence, we define the supervised loss for the representation $f$ as

$$L_{\text{Sup}}(\mathcal{T}, f) = \inf_{W \in \mathbb{R}^{K \times d}} L_{\text{Softmax}}(\mathcal{T}, Wf). \tag{10}$$

In line with the approach of [1] we analyze the supervised loss of a mean classifier [36], where for each class $c$, the rows of $W$ are set to the mean of the representations $\mu_c = \mathbb{E}_{x \sim p(\cdot|c)}[f(x)]$. We will use $L_{\text{Sup}}^{\mu}(\mathcal{T}, f)$ as shorthand for its loss. Note that $L_{\text{Sup}}^{\mu}(\mathcal{T}, f)$ is always an upper bound on $L_{\text{Sup}}(\mathcal{T}, f)$. To allow for uncertainty about the task $\mathcal{T}$, we will bound the average supervised loss for a uniform distribution $\mathcal{D}$ over $K$-way classification tasks with classes in $\mathcal{C}$.

$$L_{\text{Sup}}(f) = \mathbb{E}_{\mathcal{T} \sim \mathcal{D}} L_{\text{Sup}}(\mathcal{T}, f). \tag{11}$$

We begin by showing that the asymptotic unbiased contrastive loss is an upper bound on the supervised loss of the mean classifier.

**Lemma 4.** *For any embedding $f$, whenever $N \geq K - 1$ we have*

$$L_{\text{Sup}}(f) \leq L_{\text{Sup}}^{\mu}(f) \leq \widetilde{L}_{\text{Debiased}}^{N}(f).$$

Lemma 4 uses the asymptotic version of the debiased loss. Together with Theorem 3 and a concentration of measure result, it leads to a generalization bound for debiased contrastive learning, as we show next.

**Generalization Bound.** In practice, we use an empirical estimate $\widehat{L}_{\text{Debiased}}^{N,M}$, i.e., an average over $T$ data points $x$, with $M$ positive and $N$ negative samples for each $x$. Our algorithm learns an empirical risk minimizer $\hat{f} \in \arg\min_{f \in \mathcal{F}} \widehat{L}_{\text{Debiased}}^{N,M}(f)$ from a function class $\mathcal{F}$. The generalization depends on the *empirical Rademacher complexity* $\mathcal{R}_{\mathcal{S}}(\mathcal{F})$ of $\mathcal{F}$ with respect to our data sample $\mathcal{S} = \{x_j, x_j^+, \{u_{i,j}\}_{i=1}^{N}, \{v_{i,j}\}_{i=1}^{M}\}_{j=1}^{T}$. Let $f_{|\mathcal{S}} = (f_k(x_j), f_k(x_j^+), \{f_k(u_{i,j})\}_{i=1}^{N}, \{f_k(v_{i,j})\}_{i=1}^{M})_{j \in [T], k \in [d]} \in \mathbb{R}^{(N+M+2)dT}$ be the restriction of $f$ onto $\mathcal{S}$, using $[T] = \{1, \ldots, T\}$. Then $\mathcal{R}_{\mathcal{S}}(\mathcal{F})$ is defined as

$$\mathcal{R}_{\mathcal{S}}(\mathcal{F}) := \mathbb{E}_{\sigma} \sup_{f \in \mathcal{F}} \langle \sigma, f_{|\mathcal{S}} \rangle \tag{12}$$

where $\sigma \sim \{\pm 1\}^{(N+M+1)dT}$ are Rademacher random variables. Combining Theorem 3 and Lemma 4 with a concentration of measure argument yields the final generalization bound for debiased contrastive learning.

**Theorem 5.** *With probability at least $1 - \delta$, for all $f \in \mathcal{F}$ and $N \geq K - 1$,*

$$L_{\text{Sup}}(\hat{f}) \leq L_{\text{Debiased}}^{N,M}(f) + \mathcal{O}\left( \frac{1}{\tau^-}\sqrt{\frac{1}{N}} + \frac{\tau^+}{\tau^-}\sqrt{\frac{1}{M}} + \frac{\lambda \mathcal{R}_{\mathcal{S}}(\mathcal{F})}{T} + B\sqrt{\frac{\log\frac{1}{\delta}}{T}} \right) \tag{13}$$

*where $\lambda = \sqrt{\frac{1}{(\tau^-)^2}(\frac{M}{N} + 1) + (\tau^+)^2(\frac{N}{M} + 1)}$ and $B = \log N \left(\frac{1}{\tau^-} + \tau^+\right)$.*

The bound states that if the function class $\mathcal{F}$ is sufficiently rich to contain some embedding for which $L_{\text{Debiased}}^{N,M}$ is small, then the representation encoder $\hat{f}$, learned from a large enough dataset, will perform well on the downstream classification task. The bound also highlights the role of the positive and unlabeled sample sizes $M$ and $N$ in the objective function, in line with the observation that a larger number of negative/positive examples in the objective leads to better results [18, 2]. The last two terms in the bound grow slowly with $N$, but the effect of this on the generalization error is small if the dataset size $T$ is much larger than $N$ and $M$, as is commonly the case. The dependence on on $N$ and $T$ in Theorem 5 is roughly equivalent to the result in [1], but the two bounds are not directly comparable since the proof strategies differ.

## 7  Conclusion

In this work, we propose *debiased contrastive learning*, a new unsupervised contrastive representation learning framework that corrects for the bias introduced by the common practice of sampling negative (dissimilar) examples for a point from the overall data distribution. Our debiased objective consistently improves the state-of-the-art baselines in various benchmarks in vision, language and reinforcement learning. The proposed framework is accompanied by generalization guarantees for the downstream classification task. Interesting directions of future work include (1) trying the debiased objective in semi-supervised learning or few shot learning, and (2) studying the effect of how positive (similar) examples are drawn, e.g., analyzing different data augmentation techniques.

## Broader Impact

Unsupervised representation learning can improve learning when only small amounts of labeled data are available. This is the case in many applications of societal interest, such as medical data analysis [5, 31], the sciences [22], or drug discovery and repurposing [38]. Improving representation learning, as we do here, can potentially benefit all these applications.

However, biases in the data can naturally lead to biases in the learned representation [29]. These biases can, for example, lead to worse performance for smaller classes or groups. For instance, the majority groups are sampled more frequently than the minority ones [16]. In this respect, our method may suffer from similar biases as standard contrastive learning, and it is an interesting avenue of future research to thoroughly test and evaluate this.

**Acknowledgements**  This work was supported by MIT-IBM Watson AI Lab. an NSF TRIPODS+X grant (number: 1839258), NSF BIDGATA award 1741341 and the MIT-MSR Trustworthy and Robust AI Collaboration. We thank Wei Fang, Tongzhou Wang, Wei-Chiu Ma, and Behrooz Tahmasebi for helpful discussions and suggestions.

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
