[Supplementary Material]

# A Proofs of Theoretical Results

## A.1 Proof of Lemma 1

The first result we give shows the relation between the unbiased, and conventional (sample biased) objective.

**Lemma 1.** *For any embedding $f$ and finite $N$, we have*

$$L_{\text{Biased}}^N(f) \geq L_{\text{Unbiased}}^N(f) + \mathbb{E}_{x \sim p}\left[0 \wedge \log \frac{\mathbb{E}_{x^+ \sim p_x^+} \exp f(x)^\top f(x^+)}{\mathbb{E}_{x^- \sim p_x^-} \exp f(x)^\top f(x^-)}\right] - e^{3/2}\sqrt{\frac{\pi}{2N}}.$$

*where $a \wedge b$ denotes the minimum of two real numbers $a$ and $b$.*

*Proof.* We use the notation $h(x, \bar{x}) = \exp^{f(x)^\top f(\bar{x})}$ for the critic. We will use Theorem 3 to prove this lemma. Setting $\tau^+ = 0$, Theorem 3 states that

$$\mathbb{E}_{\substack{x \sim p \\ x^+ \sim p_x^+}}\left[-\log \frac{h(x, x^+)}{h(x, x^+) + N\mathbb{E}_{x^- \sim p}h(x, x_i^-)}\right]$$
$$- \mathbb{E}_{\substack{x \sim p \\ x^+ \sim p_x^+ \\ \{x_i^-\}_{i=1}^N \sim p^N}}\left[-\log \frac{h(x, x^+)}{h(x, x^+) + \sum_{i=1}^N h(x, x_i^-)}\right] \leq e^{3/2}\sqrt{\frac{\pi}{2N}}.$$

Equipped with this inequality, the biased objective can be decomposed into the sum of the debiased objective and a second term as follows:

$$L_{\text{Biased}}^N(f)$$
$$= \mathbb{E}_{\substack{x \sim p \\ x^+ \sim p_x^+ \\ \{x_i^-\}_{i=1}^N \sim p^N}}\left[-\log \frac{h(x, x^+)}{h(x, x^+) + \sum_{i=1}^N h(x, x_i^-)}\right]$$
$$\geq \mathbb{E}_{\substack{x \sim p \\ x^+ \sim p_x^+}}\left[-\log \frac{h(x, x^+)}{h(x, x^+) + N\mathbb{E}_{x^- \sim p_x}h(x, x^-)}\right] - e^{3/2}\sqrt{\frac{\pi}{2N}}$$
$$= \mathbb{E}_{\substack{x \sim p \\ x^+ \sim p_x^+}}\left[-\log \frac{h(x, x^+)}{h(x, x^+) + N\mathbb{E}_{x^- \sim p_x^-}h(x, x^-)}\right]$$
$$\quad + \mathbb{E}_{\substack{x \sim p \\ x^+ \sim p_x^+}}\left[\log \frac{h(x, x^+) + N\mathbb{E}_{x^- \sim p_x}h(x, x^-)}{h(x, x^+) + N\mathbb{E}_{x^- \sim p_x^-}h(x, x^-)}\right] - e^{3/2}\sqrt{\frac{\pi}{2N}}$$
$$= L_{\text{Debiased}}^N(f) + \mathbb{E}_{\substack{x \sim p \\ x^+ \sim p_x^+}}\left[\log \frac{h(x, x^+) + N\mathbb{E}_{x^- \sim p_x}h(x, x^-)}{h(x, x^+) + N\mathbb{E}_{x^- \sim p_x^-}h(x, x^-)}\right] - e^{3/2}\sqrt{\frac{\pi}{2N}}$$
$$= L_{\text{Debiased}}^N(f) + \mathbb{E}_{\substack{x \sim p \\ x^+ \sim p_x^+}}\Big[\underbrace{\log \frac{h(x, x^+) + \tau^- N\mathbb{E}_{x^- \sim p_x^-}h(x, x^-) + \tau^+ N\mathbb{E}_{x^- \sim p_x^+}h(x, x^-)}{h(x, x^+) + \tau^- N\mathbb{E}_{x^- \sim p_x^-}h(x, x^-) + \tau^+ N\mathbb{E}_{x^- \sim p_x}h(x, x^-)}}_{g(x, x^+)}\Big] - e^{3/2}\sqrt{\frac{\pi}{2N}}.$$

If $\mathbb{E}_{x^- \sim p_x^+}h(x, x^-) \geq \mathbb{E}_{x^- \sim p_x^-}h(x, x^-)$, then $g(x, x^+)$ can be lower bounded by $\log 1 = 0$. Otherwise, if $\mathbb{E}_{x^- \sim p_x^+}h(x, x^-) \leq \mathbb{E}_{x^- \sim p_x^-}h(x, x^-)$, we can use the elementary fact that $\frac{a+c}{b+c} \geq \frac{a}{b}$ for $a \leq b$ and $a, b, c \geq 0$. Combining these two cases, we conclude that

$$L_{\text{Biased}}^N(f) \geq L_{\text{Unbiased}}^N(f) + \mathbb{E}_{x \sim p}\left[0 \wedge \log \frac{\mathbb{E}_{x^+ \sim p_x^+} \exp f(x)^\top f(x^+)}{\mathbb{E}_{x^- \sim p_x^-} \exp f(x)^\top f(x^-)}\right] - e^{3/2}\sqrt{\frac{\pi}{2N}},$$

where we replaced the dummy variable $x^-$ in the numerator by $x^+$. $\qquad\square$

## A.2 Proof of Lemma 2

The next result is a consequence of the dominated convergence theorem.

**Lemma 2.** *For fixed Q and $N \to \infty$, it holds that*

$$\mathbb{E}_{\substack{x\sim p, x^+\sim p_x^+ \\ \{x_i^-\}_{i=1}^N \sim p_x^{-N}}} \left[ -\log \frac{e^{f(x)^T f(x^+)}}{e^{f(x)^T f(x^+)} + \frac{Q}{N}\sum_{i=1}^N e^{f(x)^T f(x_i^-)}} \right]$$

$$\longrightarrow \mathbb{E}_{\substack{x\sim p \\ x^+\sim p_x^+}} \left[ -\log \frac{e^{f(x)^T f(x^+)}}{e^{f(x)^T f(x^+)} + \frac{Q}{\tau^-}\left(\mathbb{E}_{x^-\sim p}[e^{f(x)^T f(x^-)}] - \tau^+\mathbb{E}_{v\sim p_x^+}[e^{f(x)^T f(v)}]\right)} \right].$$

*Proof.* Since the contrastive loss is bounded, applying the Dominated Convergence Theorem completes the proof:

$$\lim_{N\to\infty} \mathbb{E}\left[ -\log \frac{e^{f(x)^T f(x^+)}}{e^{f(x)^T f(x^+)} + \frac{Q}{N}\sum_{i=1}^N e^{f(x)^T f(x_i^-)}} \right]$$

$$= \mathbb{E}\left[ \lim_{N\to\infty} -\log \frac{e^{f(x)^T f(x^+)}}{e^{f(x)^T f(x^+)} + \frac{Q}{N}\sum_{i=1}^N e^{f(x)^T f(x_i^-)}} \right] \quad \text{(Dominated Convergence Theorem)}$$

$$= \mathbb{E}\left[ -\log \frac{e^{f(x)^T f(x^+)}}{e^{f(x)^T f(x^+)} + Q\mathbb{E}_{x^-\sim p_x^-} e^{f(x)^T f(x^-)}} \right].$$

Since $p_x^-(x') = (p(x') - \tau^+ p_x^+(x'))/\tau^-$ and by the linearity of the expectation, we have

$$\mathbb{E}_{x^-\sim p_x^-} e^{f(x)^T f(x^-)} = \tau^-\left(\mathbb{E}_{x^-\sim p}[e^{f(x)^T f(x^-)}] - \tau^+\mathbb{E}_{x^-\sim p_x^+}[e^{f(x)^T f(x^-)}]\right),$$

which completes the proof. $\qquad\square$

## A.3 Proof of Theorem 3

In order to prove Theorem 3, which shows that the empirical estimate of the asymptotic debiased objective is a good estimate, we first seek a bound on the tail probability that the difference between the integrands of the asymptotic and non-asymptotic objective functions i slarge. That is, we wish to bound the probability that the following quantity is greater than $\varepsilon$:

$$\Delta = \left| -\log \frac{h(x, x^+)}{h(x, x^+) + Qg(x, \{u_i\}_{i=1}^N, \{v_i\}_{i=1}^M)} + \log \frac{h(x, x^+)}{h(x, x^+) + Q\mathbb{E}_{x^-\sim p_x^-} h(x, x^-)} \right|,$$

where we again write $h(x, \bar{x}) = \exp^{f(x)^\top f(\bar{x})}$. Note that implicitly, $\Delta$ depends on $x, x^+$ and the collections $\{u_i\}_{i=1}^N$ and $\{v_i\}_{i=1}^M$. We achieve control over the tail via the following lemma.

**Lemma A.2.** *Let $x$ and $x^+$ in $\mathcal{X}$ be fixed. Further, let $\{u_i\}_{i=1}^N$ and $\{v_i\}_{i=1}^M$ be collections of i.i.d. random variables sampled from $p$ and $p_x^+$ respectively. Then for all $\varepsilon > 0$,*

$$\mathbb{P}(\Delta \geq \varepsilon) \leq 2\exp\left(-\frac{N\varepsilon^2(\tau^-)^2}{2e^3}\right) + 2\exp\left(-\frac{M\varepsilon^2(\tau^-/\tau^+)^2}{2e^3}\right).$$

We delay the proof until after we prove Theorem 3, which we are ready to prove with this fact in hand.

**Theorem 3.** *For any embedding $f$ and finite $N$ and $M$, we have*

$$\left| \widetilde{L}_{\text{Debiased}}^N(f) - L_{\text{Debiased}}^{N,M}(f) \right| \leq \frac{e^{3/2}}{\tau^-}\sqrt{\frac{\pi}{2N}} + \frac{e^{3/2}\tau^+}{\tau^-}\sqrt{\frac{\pi}{2M}}.$$

*Proof.* By Jensen's inequality, we may push the absolute value inside the expectation to see that $|\widetilde{L}_{\text{Unbiased}}^N(f) - L_{\text{Debiased}}^{N,M}(f)| \leq \mathbb{E}\Delta$. All that remains is to exploit the exponential tail bound of Lemma $A.2$.

To do this we write the expectation of $\Delta$ for fixed $x, x^+$ as the integral of its tail probability,

$$\mathbb{E}\,\Delta = \mathbb{E}_{x,x^+}\left[\mathbb{E}[\Delta|x,x^+]\right] = \mathbb{E}_{x,x^+}\left[\int_0^\infty \mathbb{P}(\Delta \geq \varepsilon|x,x^+)\mathrm{d}\varepsilon\right]$$

$$\leq \int_0^\infty 2\exp\left(-\frac{N\varepsilon^2(\tau^-)^2}{2e^3}\right)\mathrm{d}\varepsilon + \int_0^\infty 2\exp\left(-\frac{M\varepsilon^2(\tau^-/\tau^+)^2}{2e^3}\right)\mathrm{d}\varepsilon.$$

The outer expectation disappears since the tail probably bound of Theorem A.2 holds uniformly for all fixed $x, x^+$. Both integrals can be computed analytically using the classical identity

$$\int_0^\infty e^{-cz^2}\mathrm{d}z = \frac{1}{2}\sqrt{\frac{\pi}{c}}.$$

Applying the identity to each integral we finally obtain the claimed bound,

$$\sqrt{\frac{2e^3\pi}{(\tau^-)^2 N}} + \sqrt{\frac{2e^3\pi}{(\tau^-/\tau^+)^2 M}} = \frac{e^{3/2}}{\tau^-}\sqrt{\frac{2\pi}{N}} + \frac{e^{3/2}\tau^+}{\tau^-}\sqrt{\frac{2\pi}{M}}.$$

$\square$

We still owe the reader a proof of Lemma A.2, which we give now.

*Proof of Lemma A.2.* We first decompose the probability as

$$\mathbb{P}\left(\left| -\log\frac{h(x,x^+)}{h(x,x^+) + Qg(x,\{u_i\}_{i=1}^N,\{v_i\}_{i=1}^M)} + \log\frac{h(x,x^+)}{h(x,x^+) + Q\mathbb{E}_{x^-\sim p_x^-}h(x,x^-)}\right| \geq \varepsilon\right)$$

$$= \mathbb{P}\left(\left|\log\left\{h(x,x^+) + Qg(x,\{u_i\}_{i=1}^N,\{v_i\}_{i=1}^M)\right\} - \log\left\{h(x,x^+) + Q\mathbb{E}_{x^-\sim p_x^-}h(x,x^-)\right\}\right| \geq \varepsilon\right)$$

$$= \mathbb{P}\left(\log\left\{h(x,x^+) + Qg(x,\{u_i\}_{i=1}^N,\{v_i\}_{i=1}^M)\right\} - \log\left\{h(x,x^+) + Q\mathbb{E}_{x^-\sim p_x^-}h(x,x^-)\right\} \geq \varepsilon\right)$$

$$+ \mathbb{P}\left(-\log\left\{h(x,x^+) + Qg(x,\{u_i\}_{i=1}^N,\{v_i\}_{i=1}^M)\right\} + \log\left\{h(x,x^+) + Q\mathbb{E}_{x^-\sim p_x^-}h(x,x^-)\right\} \geq \varepsilon\right)$$

where the final equality holds simply because $|X| \geq \varepsilon$ if and only if $X \geq \varepsilon$ or $-X \geq \varepsilon$. The first term can be bounded as

$$\mathbb{P}\left(\log\left\{h(x,x^+) + Qg(x,\{u_i\}_{i=1}^N,\{v_i\}_{i=1}^M)\right\} - \log\left\{h(x,x^+) + Q\mathbb{E}_{x^-\sim p_x^-}h(x,x^-)\right\} \geq \varepsilon\right)$$

$$= \mathbb{P}\left(\log\frac{h(x,x^+) + Qg(x,\{u_i\}_{i=1}^N,\{v_i\}_{i=1}^M)}{h(x,x^+) + Q\mathbb{E}_{x^-\sim p_x^-}h(x,x^-)} \geq \varepsilon\right)$$

$$\leq \mathbb{P}\left(\frac{Qg(x,\{u_i\}_{i=1}^N,\{v_i\}_{i=1}^M) - Q\mathbb{E}_{x^-\sim p_x^-}h(x,x^-)}{h(x,x^+) + Q\mathbb{E}_{x^-\sim p_x^-}h(x,x^-)} \geq \varepsilon\right)$$

$$= \mathbb{P}\left(g(x,\{u_i\}_{i=1}^N,\{v_i\}_{i=1}^M) - \mathbb{E}_{x^-\sim p_x^-}h(x,x^-) \geq \varepsilon\left\{\frac{1}{Q}h(x,x^+) + \mathbb{E}_{x^-\sim p_x^-}h(x,x^-)\right\}\right)$$

$$\leq \mathbb{P}\left(g(x,\{u_i\}_{i=1}^N,\{v_i\}_{i=1}^M) - \mathbb{E}_{x^-\sim p_x^-}h(x,x^-) \geq \varepsilon e^{-1}\right). \tag{14}$$

The first inequality follows by applying the fact that $\log x \leq x - 1$ for $x > 0$. The second inequality holds since $\frac{1}{Q}h(x,x^+) + \mathbb{E}_{x^-\sim p_x^-}h(x,x^-) \geq 1/e$. Next, we move on to bounding the second term,

which proceeds similarly, using the same two bounds.

$$\mathbb{P}\Bigg\{-\log\big(h(x,x^+)+Qg(x,\{u_i\}_{i=1}^N,\{v_i\}_{i=1}^M)\big)+\log\big\{h(x,x^+)+Q\mathbb{E}_{x^-\sim p_x^-}h(x,x^-)\big\}\geq\varepsilon\Bigg)$$

$$=\mathbb{P}\Bigg(\log\frac{h(x,x^+)+Q\mathbb{E}_{x^-\sim p_x^-}h(x,x^-)}{h(x,x^+)+Qg(x,\{u_i\}_{i=1}^N,\{v_i\}_{i=1}^M)}\geq\varepsilon\Bigg)$$

$$\leq\mathbb{P}\Bigg(\frac{Q\mathbb{E}_{x^-\sim p_x^-}h(x,x^-)-Qg(x,\{u_i\}_{i=1}^N,\{v_i\}_{i=1}^M)}{h(x,x^+)+Qg(x,\{u_i\}_{i=1}^N,\{v_i\}_{i=1}^M)}\geq\varepsilon\Bigg)$$

$$=\mathbb{P}\Bigg(\mathbb{E}_{x^-\sim p_x^-}h(x,x^-)-g(x,\{u_i\}_{i=1}^N,\{v_i\}_{i=1}^M)\geq\varepsilon\Big\{\frac{1}{Q}h(x,x^+)+g(x,\{u_i\}_{i=1}^N,\{v_i\}_{i=1}^M)\Big\}\Bigg)$$

$$\leq\mathbb{P}\Bigg(\mathbb{E}_{x^-\sim p_x^-}h(x,x^-)-g(x,\{u_i\}_{i=1}^N,\{v_i\}_{i=1}^M)\geq\varepsilon e^{-1}\Bigg). \tag{15}$$

Combining equation (14) and equation (15), we have

$$\mathbb{P}(\Delta\geq\varepsilon)\leq\mathbb{P}\Bigg(\big|g(x,\{u_i\}_{i=1}^N,\{v_i\}_{i=1}^M)-\mathbb{E}_{x^-\sim p_x^-}h(x,x^-)\big|\geq\varepsilon e^{-1}\Bigg).$$

We then proceed to bound the right hand tail probability. We are bounding the tail of a difference of the form $|\max(a,b)-c|$ where $c\geq b$. Notice that $|\max(a,b)-c|\leq|a-c|$. If $a>b$ then this relation is obvious, while if $a\leq b$ we have $|\max(a,b)-c|=|b-c|=c-b\leq c-a\leq|a-c|$. Using this elementary observation, we can decompose the random variable whose tail we wish to control as follows:

$$\big|g(x,\{u_i\}_{i=1}^N,\{v_i\}_{i=1}^M)-\mathbb{E}_{x^-\sim p_x^-}h(x,x^-)\big|$$

$$\leq\frac{1}{\tau^-}\Bigg|\frac{1}{N}\sum_{i=1}^N\mathbb{E}_{x\sim p}h(x,u_i)-\mathbb{E}_{\substack{x^-\sim p\\x\sim p}}h(x,x^-)\Bigg|+\frac{\tau^+}{\tau^-}\Bigg|\frac{1}{M}\sum_{i=1}^M\mathbb{E}_{x\sim p}h(x,v_i)-\mathbb{E}_{\substack{x^-\sim p_x^+\\x\sim p}}h(x,x^-)\Bigg|$$

Using this observation, we find that

$$\mathbb{P}\Bigg(\big|g(x,\{u_i\}_{i=1}^N,\{v_i\}_{i=1}^M)-\mathbb{E}_{x^-\sim p_x^-}h(x,x^-)\big|\geq\varepsilon e^{-1}\Bigg)$$

$$\leq\mathbb{P}\Bigg(\Big|\frac{1}{\tau^-}\Big(\frac{1}{N}\sum_{i=1}^N e^{f(x)^T f(u_i)}-\tau^+\frac{1}{M}\sum_{i=1}^M e^{f(x)^T f(v_i)}\Big)-\mathbb{E}_{x^-\sim p_x^-}h(x,x^-)\Big|\geq\varepsilon e^{-1}\Bigg)$$

$$\leq\mathrm{I}(\varepsilon)+\mathrm{II}(\varepsilon).$$

where

$$\mathrm{I}(\varepsilon)=\mathbb{P}\Bigg(\frac{1}{\tau^-}\Big|\frac{1}{N}\sum_{i=1}^N h(x,u_i)-\mathbb{E}_{x^-\sim p}h(x,x^-)\Big|\geq\frac{\varepsilon e^{-1}}{2}\Bigg)$$

$$\mathrm{II}(\varepsilon)=\mathbb{P}\Bigg(\frac{\tau^+}{\tau^-}\Big|\frac{1}{M}\sum_{i=1}^M h(x,v_i)-\mathbb{E}_{x^-\sim p_x^+}h(x,x^-)\Big|\geq\frac{\varepsilon e^{-1}}{2}\Bigg).$$

Hoeffding's inequality states that if $X,X_1,\ldots,X_N$ are i.i.d random variables bounded in the range $[a,b]$, then

$$\mathbb{P}\Bigg(\Big|\frac{1}{n}\sum_{i=1}^N X_i-\mathbb{E}X\Big|\geq\varepsilon\Bigg)\leq 2\exp\Big(-\frac{2N\varepsilon^2}{b-a}\Big).$$

In our particular case, $e^{-1}\leq h(x,\bar{x})\leq e$, yielding the following bound on the tails of both terms:

$$\mathrm{I}(\varepsilon)\leq 2\exp\Big(-\frac{N\varepsilon^2(\tau^-)^2}{2e^3}\Big)\quad\text{and}\quad\mathrm{II}(\varepsilon)\leq 2\exp\Big(-\frac{M\varepsilon^2(\tau^-/\tau^+)^2}{2e^3}\Big).$$

$\square$

## A.4 Proof of Lemma 4

**Lemma 4.** *For any embedding $f$, whenever $N \geq K - 1$ we have*

$$L_{\text{Sup}}(f) \leq L_{\text{Sup}}^{\mu}(f) \leq \widetilde{L}_{\text{Debiased}}^{N}(f).$$

*Proof.* We first show that $N = K - 1$ gives the smallest loss:

$$\widetilde{L}_{\text{Unbiased}}^{N}(f) = \mathop{\mathbb{E}}_{\substack{x \sim p \\ x^+ \sim p_x^+}} \left[ -\log \frac{e^{f(x)^T f(x^+)}}{e^{f(x)^T f(x^+)} + N\mathbb{E}_{x^- \sim p_x^-} e^{f(x)^T f(x^-)}} \right]$$

$$\geq \mathop{\mathbb{E}}_{\substack{x \sim p \\ x^+ \sim p_x^+}} \left[ -\log \frac{e^{f(x)^T f(x^+)}}{e^{f(x)^T f(x^+)} + (K-1)\mathbb{E}_{x^- \sim p_x^-} e^{f(x)^T f(x^-)}} \right]$$

$$= L_{\text{Unbiased}}^{K-1}(f)$$

To show that $L_{\text{Unbiased}}^{K-1}(f)$ is an upper bound on the supervised loss $L_{\sup}(f)$, we additionally introduce a task specific class distribution $\rho_{\mathcal{T}}$ which is a uniform distribution over all the possible $K$-way classification tasks with classes in $\mathcal{C}$. That is, we consider all the possible task with $K$ distinct classes $\{c_1, \ldots, c_K\} \subseteq \mathcal{C}$.

$$L_{\text{Unbiased}}^{K-1}(f)$$

$$= \mathop{\mathbb{E}}_{\substack{x \sim p \\ x^+ \sim p_x^+}} \left[ -\log \frac{e^{f(x)^T f(x^+)}}{e^{f(x)^T f(x^+)} + (K-1)\mathbb{E}_{x^- \sim p_x^-} e^{f(x)^T f(x^-)}} \right]$$

$$= \mathop{\mathbb{E}}_{\mathcal{T} \sim \mathcal{D}} \mathop{\mathbb{E}}_{\substack{c \sim \rho_{\mathcal{T}}; x \sim p(\cdot|c) \\ x^+ \sim p(\cdot|c)}} \left[ -\log \frac{e^{f(x)^T f(x^+)}}{e^{f(x)^T f(x^+)} + (K-1)\mathbb{E}_{\mathcal{T} \sim \mathcal{D}} \mathbb{E}_{\rho_{\mathcal{T}}(c^- \sim |c^- \neq h(x))} \mathbb{E}_{x^- \sim p(\cdot|c^-)} e^{f(x)^T f(x^-)}} \right]$$

$$\geq \mathop{\mathbb{E}}_{\mathcal{T} \sim \mathcal{D}} \mathop{\mathbb{E}}_{c \sim \rho_{\mathcal{T}}; x \sim p(\cdot|c)} \left[ -\log \frac{e^{f(x)^T \mathbb{E}_{x^+ \sim p(\cdot|c)} f(x^+)}}{e^{f(x)^T \mathbb{E}_{x^+ \sim p_{x,\mathcal{T}}^+} f(x^+)} + (K-1)\mathbb{E}_{\mathcal{T} \sim \mathcal{D}} \mathbb{E}_{\rho_{\mathcal{T}}(c^-|c^- \neq h(x))} \mathbb{E}_{x^- \sim p(\cdot|c^-)} e^{f(x)^T f(x^-)}} \right]$$

$$\geq \mathop{\mathbb{E}}_{\mathcal{T} \sim \mathcal{D}} \mathop{\mathbb{E}}_{c \sim \rho_{\mathcal{T}}; x \sim p(\cdot|c)} \left[ -\log \frac{e^{f(x)^T \mathbb{E}_{x^+ \sim p(\cdot|c)} f(x^+)}}{e^{f(x)^T \mathbb{E}_{x^+ \sim p(\cdot|c)} f(x^+)} + (K-1)\mathbb{E}_{\rho_{\mathcal{T}}(c^-|c^- \neq h(x))} \mathbb{E}_{x^- \sim p(\cdot|c^-)} e^{f(x)^T f(x^-)}} \right]$$

$$= \mathop{\mathbb{E}}_{\mathcal{T} \sim \mathcal{D}} \mathop{\mathbb{E}}_{c \sim \rho_{\mathcal{T}}; x \sim p(\cdot|c)} \left[ -\log \frac{e^{f(x)^T \mathbb{E}_{x^+ \sim p(\cdot|c)} f(x^+)}}{e^{f(x)^T \mathbb{E}_{x^+ \sim p(\cdot|c)} f(x^+)} + (K-1)\mathbb{E}_{\rho_{\mathcal{T}}(c^-|c^- \neq h(x))} \mathbb{E}_{x^- \sim p(\cdot|c^-)} e^{f(x)^T f(x^-)}} \right]$$

$$\geq \mathop{\mathbb{E}}_{\mathcal{T} \sim \mathcal{D}} \mathop{\mathbb{E}}_{c \sim \rho_{\mathcal{T}}; x \sim p(\cdot|c)} \left[ -\log \frac{e^{f(x)^T \mathbb{E}_{x^+ \sim p(\cdot|c)} f(x^+)}}{e^{f(x)^T \mathbb{E}_{x^+ \sim p(\cdot|c)} f(x^+)} + (K-1)\mathbb{E}_{\rho_{\mathcal{T}}(c^-|c^- \neq h(x))} e^{f(x)^T \mathbb{E}_{x^- \sim p(\cdot|c^-)} f(x^-)}} \right]$$

$$= \mathop{\mathbb{E}}_{\mathcal{T} \sim \mathcal{D}} \mathop{\mathbb{E}}_{c \sim \rho_{\mathcal{T}}; x \sim p(\cdot|c)} \left[ -\log \frac{\exp(f(x)^T \mu_c)}{\exp(f(x)^T \mu_c) + \sum_{c^- \in \mathcal{T}, c^- \neq c} \exp(f(x)^T \mu_{c^-})} \right]$$

$$= \mathbb{E}_{\mathcal{T} \sim \mathcal{D}} L_{\text{Sup}}^{\mu}(\mathcal{T}, f)$$

$$= \bar{L}_{\text{Sup}}^{\mu}(f)$$

where the three inequalities follow from Jensen's inequality. The first and third inequality shift the expectations $\mathbb{E}_{x^+ \sim p_{x,\mathcal{T}}^+}$ and $\mathbb{E}_{x^- \sim p(\cdot|c^-)}$, respectively, via the convexity of the functions and the second moves the expectation $\mathbb{E}_{\mathcal{T} \sim \mathcal{D}}$ out using concavity. Note that $\bar{L}_{\text{Sup}}(f) \leq \bar{L}_{\text{Sup}}^{\mu}(f)$ holds trivially. $\square$

## A.5 Proof of Theorem 5

We wish to derive a data dependent bound on the downstream supervised generalization error of the debiased contrastive objective. Recall that a sample $(x, x^+, \{u_i\}_{i=1}^N, \{v_i\}_{i=1}^M)$ yields loss

$$-\log\left\{\frac{e^{f(x)^\top f(x^+)}}{e^{f(x)^\top f(x^+)} + Ng(x, \{u_i\}_{i=1}^N, \{v_i\}_{i=1}^M)}\right\} = \log\left\{1 + N\frac{g(x, \{u_i\}_{i=1}^N, \{v_i\}_{i=1}^M)}{e^{f(x)^\top f(x^+)}}\right\}$$

which is equal to $\ell\left(\left\{f(x)^\top\left(f(u_i) - f(x^+)\right)\right\}_{i=1}^N, \left\{f(x)^\top\left(f(v_i) - f(x^+)\right)\right\}_{i=1}^M\right)$, where we define

$$\ell(\{a_i\}_{i=1}^N, \{b_i\}_{i=1}^M) = \log\left\{1 + N\max\left(\frac{1}{\tau^-}\frac{1}{N}\sum_{i=1}^N a_i - \tau^+\frac{1}{M}\sum_{i=1}^M b_i, e^{-1}\right)\right\}.$$

To derive our bound, we will exploit a concentration of measure result due to [1]. They consider an objective of the form

$$L_{un}(f) = \mathbb{E}\left[\ell(\{f(x)^\top\left(f(x_i) - f(x^+)\right)\}_{i=1}^k)\right],$$

where $(x, x^+, x_1^-, \ldots, x_k^-)$ are sampled from any fixed distribution on $\mathcal{X}^{k+2}$ (they were particularly focused on the case where $x_i^- \sim p$, but the proof holds for arbitrary distributions). Let $\mathcal{F}$ be a class of representation functions $\mathcal{X} \to \mathbb{R}^d$ such that $\|f(\cdot)\| \leq R$ for $R > 0$. The corresponding empirical risk minimizer is

$$\hat{f} \in \arg\min_{f \in \mathcal{F}} \frac{1}{T}\sum_{j=1}^T \ell\left(\{f(x_j)^\top\left(f(x_{ji}) - f(x^+)\right)\}_{i=1}^k\right)$$

over a training set $\mathcal{S} = \{(x_j, x_j^+, x_{j1}^-, \ldots, x_{jk}^-)\}_{j=1}^T$ of i.i.d. samples. Their result bounds the loss of the empirical risk minimizer as follows.

**Lemma A.3.** [1] *Let $\ell : \mathbb{R}^k \to \mathbb{R}$ be $\eta$-Lipschitz and bounded by $B$. Then with probability at least $1 - \delta$ over the training set $\mathcal{S} = \{(x_j, x_j^+, x_{j1}^-, \ldots, x_{jk}^-)\}_{j=1}^T$, for all $f \in \mathcal{F}$*

$$L_{un}(\hat{f}) \leq L_{un}(f) + \mathcal{O}\left(\frac{\eta R\sqrt{k}\mathcal{R}_\mathcal{S}(\mathcal{F})}{T} + B\sqrt{\frac{\log\frac{1}{\delta}}{T}}\right)$$

*where*

$$\mathcal{R}_\mathcal{S}(\mathcal{F}) = \mathbb{E}_{\sigma \sim \{\pm 1\}^{(k+2)dT}}\left[\sup_{f \in \mathcal{F}}\langle\sigma, f_{|\mathcal{S}}\rangle\right],$$

*and $f_{|\mathcal{S}} = \left(f_t(x_j), f_t(x_j^+), f_t(x_{j1}^-), \ldots, f_t(x_{jk}^-)\right)_{\substack{j \in [T] \\ t \in [d]}}$.*

In our context, we have $k = N + M$ and $R = e$. So, it remains to obtain constants $\eta$ and $B$ such that $\ell(\{a_i\}_{i=1}^N, \{b_i\}_{i=1}^M)$ is $\eta$-Lipschitz, and bounded by $B$. Note that since we consider normalized embeddings $f$, we have $\|f(\cdot)\| \leq 1$ and therefore only need to consider the domain where $e^{-1} \leq a_i, b_i \leq e$.

**Lemma A.4.** *Suppose that $e^{-1} \leq a_i, b_i \leq e$. The function $\ell(\{a_i\}_{i=1}^N, \{b_i\}_{i=1}^M)$ is $\eta$-Lipschitz, and bounded by $B$ for*

$$\eta = e \cdot \sqrt{\frac{1}{(\tau^-)^2 N} + \frac{(\tau^+)^2}{M}}, \qquad B = \mathcal{O}\left(\log N\left(\frac{1}{\tau^-} + \tau^+\right)\right).$$

*Proof.* First, it is easily observed that $\ell$ is upper bounded by plugging in $a_i = e$ and $b_i = e^{-1}$, yielding a bound of

$$\log\left\{1 + N\max\left(\frac{1}{\tau^-}e - \tau^+ e^{-1}, e^{-1}\right)\right\} = \mathcal{O}\left(\log N\left(\frac{1}{\tau^-} + \tau^+\right)\right).$$

To bound the Lipschitz constant we view $\ell$ as a composition $\ell(\{a_i\}_{i=1}^N, \{b_i\}_{i=1}^M) = \phi\left(g\left(\ell(\{a_i\}_{i=1}^N, \{b_i\}_{i=1}^M)\right)\right)$ where[1],

$$\phi(z) = \log\left(1 + N\max(z, e^{-1})\right)$$

$$g(\{a_i\}_{i=1}^N, \{b_i\}_{i=1}^M) = \frac{1}{\tau^-}\frac{1}{N}\sum_{i=1}^N a_i - \tau^+\frac{1}{M}\sum_{i=1}^M b_i.$$

If $z < e^{-1}$ then $\partial_z\phi(z) = 0$, while if $z \geq e^{-1}$ then $\partial_z\phi(z) = \frac{N}{1+Nz} \leq \frac{N}{1+Ne^{-1}} \leq e$. We therefore conclude that $\phi$ is $e$-Lipschitz. Meanwhile, $\partial_{a_i}g = \frac{1}{\tau^-N}$ and $\partial_{b_i}g = \frac{\tau^+}{M}$. The Lipschitz constant of $g$ is bounded by the Forbenius norm of the Jacobian of $g$, which equals

$$\sqrt{\sum_{i=1}^N \frac{1}{(\tau^-N)^2} + \sum_{j=1}^M \frac{(\tau^+)^2}{M^2}} = \sqrt{\frac{1}{(\tau^-)^2N} + \frac{(\tau^+)^2}{M}}.$$

$\square$

Now we have control on the bound on $\ell$ and its Lipschitz constant, we are ready to prove Theorem 5 by combining several of our previous results with Lemma A.3.

**Theorem 5.** *With probability at least $1 - \delta$, for all $f \in \mathcal{F}$ and $N \geq K - 1$,*

$$L_{\text{Sup}}(\hat{f}) \leq L_{\text{Sup}}^\mu(f) \leq L_{\text{Debiased}}^{N,M}(f) + \mathcal{O}\left(\frac{1}{\tau^-}\sqrt{\frac{1}{N}} + \frac{\tau^+}{\tau^-}\sqrt{\frac{1}{M}} + \frac{\lambda\mathcal{R}_\mathcal{S}(\mathcal{F})}{T} + B\sqrt{\frac{\log\frac{1}{\delta}}{T}}\right)$$

*where $\lambda = \sqrt{\frac{1}{\tau^{-2}}(\frac{M}{N}+1) + \tau^{+2}(\frac{N}{M}+1)}$ and $B = \log N\left(\frac{1}{\tau^-} + \tau^+\right)$.*

*Proof.* By Lemma 4 and Theorem 3 we have

$$L_{\text{sup}}(\hat{f}) \leq \widetilde{L}_{\text{Unbiased}}^N(\hat{f}) \leq L_{\text{Debiased}}^{N,M}(\hat{f}) + \frac{e^{3/2}}{\tau^-}\sqrt{\frac{\pi}{2N}} + \frac{e^{3/2}\tau^+}{\tau^-}\sqrt{\frac{\pi}{2M}}.$$

Combining Lemma A.3 and Lemma A.4, with probability at least $1 - \delta$, for all $f \in \mathcal{F}$, we have

$$L_{\text{Debiased}}^{N,M}(\hat{f}) \leq L_{\text{Debiased}}^{N,M}(f) + \mathcal{O}\left(\frac{\lambda\mathcal{R}_\mathcal{S}(\mathcal{F})}{T} + B\sqrt{\frac{\log\frac{1}{\delta}}{T}}\right),$$

where $\lambda = \eta\sqrt{k} = \sqrt{\frac{1}{\tau^{-2}}(\frac{M}{N}+1) + \tau^{+2}(\frac{N}{M}+1)}$ and $B = \log N\left(\frac{1}{\tau^-} + \tau^+\right)$. $\square$

### A.6 Derivation of Equation (4)

In Section 4, we mentioned that the obvious way to approximate the unbiased objective is to replace $p_x^-$ with $p_x^-(x') = (p(x') - \tau^+p_x^+(x'))/\tau^-$ and then use the empirical counterparts for $p$ and $p_x^+$, and that this yields an objective that is a sum of $N + 1$ expectations. To give the derivation of this claim, let

$$\ell(x, x^+, \{x_i^-\}_{i=1}^N, f) = -\log\frac{e^{f(x)^Tf(x^+)}}{e^{f(x)^Tf(x^+)} + \sum_{i=1}^N e^{f(x)^Tf(x_i^-)}}.$$

We plug in the decomposition as follows:

$$\mathbb{E}_{\substack{x\sim p, x^+\sim p_x^+ \\ \{x_i^-\}_{i=1}^N\sim p_x^-}}[\ell(x, x^+, \{x_i^-\}_{i=1}^N, f)]$$

$$= \int p(x)p_x^+(x^+)\prod_{i=1}^N p_x^-(x_i^-)\ell(x, x^+, \{x_i^-\}_{i=1}^N, f)\mathrm{d}x\mathrm{d}x^+\prod_{i=1}^N \mathrm{d}x_i^-$$

$$= \int p(x)p_x^+(x^+)\prod_{i=1}^N \frac{p(x_i^-)-\tau^+ p_x^+(x_i^-)}{\tau^-}\ell(x, x^+, \{x_i^-\}_{i=1}^N, f)\mathrm{d}x\mathrm{d}x^+\prod_{i=1}^N \mathrm{d}x_i^-$$

$$= \frac{1}{(\tau^-)^N}\int p(x)p_x^+(x^+)\prod_{i=1}^N \left(p(x_i^-)-\tau^+ p_x^+(x_i^-)\right)\ell(x, x^+, \{x_i^-\}_{i=1}^N, f)\mathrm{d}x\mathrm{d}x^+\prod_{i=1}^N \mathrm{d}x_i^-.$$

By the Binomial Theorem, the product inside the integral can be separated into $N+1$ groups corresponding to how many $x_i^-$ are sampled from $p$.

$$(1)\qquad \prod_{i=1}^N p(x_i^-)$$

$$(2)\qquad \binom{N}{1}(-\tau^+)p_x^+(x_1^-)\prod_{i=2}^N p(x_i^-)$$

$$(3)\qquad \binom{N}{2}\prod_{j=1}^2(-\tau^+)p_x^+(x_j^-)\prod_{i=3}^N p(x_i^-)$$

$$\ldots$$

$$(k+1)\qquad \binom{N}{k}\prod_{j=1}^k(-\tau^+)p_x^+(x_j^-)\prod_{i=k+1}^N p(x_i^-)$$

$$\ldots$$

$$(N+1)\qquad \prod_{i=1}^N(-\tau^+)p_x^+(x_i^-)$$

In particular, the objective becomes

$$\frac{1}{(\tau^-)^N}\sum_{k=0}^N\binom{N}{k}(-\tau^+)^k\mathbb{E}_{\substack{x\sim p, x^+\sim p_x^+ \\ \{x_i^-\}_{i=1}^k\sim p_x^+ \\ \{x_i^-\}_{i=k+1}^N\sim p}}\left[-\log\frac{e^{f(x)^T f(x^+)}}{e^{f(x)^T f(x^+)}+\sum_{i=1}^N e^{f(x)^T f(x_i^-)}}\right],$$

where $\{x_i^-\}_{i=k}^j=\emptyset$ if $k>j$. Note that this is exactly the *Inclusion–exclusion principle*. The numerical value of this objective is extremely small when $N$ is large. We tried various approaches to optimize this objective, but none of them worked.

## B  Experimental Details

**CIFAR10 and STL10**   We adopt PyTorch to implement SimCLR [2] with ResNet-50 [17] as the encoder architecture and use the Adam optimizer [23] with learning rate $0.001$ and weight decay $1e-6$. We set the temperature $t$ to $0.5$ and the dimension of the latent vector to $128$. All the models are trained for $400$ epochs. The data augmentation uses the following PyTorch code:

The models are evaluated by training a linear classifier with cross entropy loss after fixing the learned embedding. We again use the Adam optimizer with learning rate $0.001$ and weight decay $1e-6$.

**Imagenet-100**   We adopt the official code[2] for contrastive multiview coding (CMC) [40]. To implement the debiased objective, we only modify the "NCE/NCECriterion.py" file and adopt the rest

```
1  train_transform = transforms.Compose([
2      transforms.RandomResizedCrop(32),
3      transforms.RandomHorizontalFlip(p=0.5),
4      transforms.RandomApply([transforms.ColorJitter(0.4, 0.4, 0.4, 0.1)], p
       =0.8),
5      transforms.RandomGrayscale(p=0.2),
6      GaussianBlur(kernel_size=int(0.1 * 32)),
7      transforms.ToTensor(),
8      transforms.Normalize([0.4914, 0.4822, 0.4465], [0.2023, 0.1994, 0.2010])
       ])
```

Figure 6: PyTorch code for SimCLR data augmentation.

of the code without change. The temperature of CMC is set to $0.07$, which often makes the estimator $\frac{1}{\tau^-}\left(\frac{1}{N}\sum_{i=1}^{N}e^{f(x)^T f(u_i)} - \tau^+ \frac{1}{M}\sum_{i=1}^{M}e^{f(x)^T f(v_i)}\right)$ less than $e^{-1/t}$. To retain the learning signal, whenever the estimator is less than $e^{-1/t}$, we optimize the biased loss instead. This improves the convergence and stability of our method.

**Sentence Embedding** We adopt the official code[3] for quick-thought (QT) vectors [28]. To implement the debiased objective, we only modify the "src/s2v-model.py" file and adopt the rest of the code without changes. Since the official BookCorpus [25] dataset is missing, we use the inofficial version[4] for the experiments. The feature vector of QT is not normalized, therefore, we simply constrain the estimator described in equation (7) to be greater than zero.

**Reinforcement Learning** We adopt the official code[5] of Contrastive unsupervised representations for reinforcement learning (CURL) [37]. To implement the debiased objective, we only modify the "curl-sac.py" file and adopt the rest of the code without changes. We again constrain the estimator described in equation (7) to be greater than zero since the feature vector of CURL is not normalized.

## Footnotes

[1]Note the definition of $g$ is slightly modified in this context.

[2]https://github.com/HobbitLong/CMC/

[3]https://github.com/lajanugen/S2V

[4]https://github.com/soskek/bookcorpus

[5]https://github.com/MishaLaskin/curl