[Reviews · NeurIPS 2020]

Review 1

Summary and Contributions: The paper is on contrastive representation learning. Current implementations of contrastive learning require sampling negatives. For convenience, people typically sample negatives from a uniform distribution over the unlabeled dataset. This could potentially sample images from the same class / category as the anchor especially one trains with really large batch sizes that are commonly and beneficial to contrastive learning. The paper proposes a mathematically motivated fix to this bias in the contrastive learning objective with a practical way to implement the fix, showing good gains on multiple benchmarks in vision, NLP, RL.

Strengths: The paper is well written and the motivation is explained well, both mathematically and with a visual illustration. It is theoretically grounded and is also trying to address a practical problem for a very important and currently a topic of huge interest to the community (contrastive learning). The empirical evaluations are clean and well ablated. It is quite significant / novel in the sense that while multiple papers are published on contrastive learning in terms of modifying the architecture / computation scale, not much focus is on the negatives. They point out a problem that definitely exists in the way we sample negatives (potential positives being used as negatives), propose a mathematical fix, and an empirical trick that can approximate it. They ablate for the design choices of the weighting constant as well as the number of positives in the new denominator used for the contrastive loss. The code is also open sourced and was clean to follow. The improvements on STL-10, CIFAR, NLP and RL benchmarks are quite clear and significant. Given the wide interest in contrastive learning and its applications to various fields as well as the big interest in unsupervised representation learning, this paper has interesting theoretical as well as practical insights that will be of interest to a wide audience in NeurIPS and should be accepted.

Weaknesses: 1. The paper could do a better job at citing relevant work on contrastive learning, eg, papers like CPCv2, MoCov2 are not cited. 2. The paper could point some of the things to be improved, such as the need to tweak the tau+ parameter in case you just use one positive sample, or the need to do forward props of the multiple positive samples for a fixed tau+. It is also unclear how to pick a tau+ and M (number of positives in the denominator) for a new domain, how to scale it for large batch sizes, etc. It would be nice if the paper can mention these as areas of improvement (even if they dont have the compute resources to study these themselves).

Correctness: To my knowledge, yes.

Clarity: Yes.

Relation to Prior Work: Yes.

Reproducibility: Yes

Additional Feedback:


Review 2

Summary and Contributions: This paper studies the contrastive loss for representation learning used by a couple of papers. The authors claim that the loss is biased and can be corrected with slight modification and include theoretical analysis. The new unbiased loss is compared with the original on image representation benchmarks (Cifar10, STL10, ImageNet100), sentence representation learning benchmark and an RL benchmark.

Strengths: - The authors propose an interesting idea to improve a commonly used representation learning loss function - The paper is well written - The method is well theoretical grounded - Reasonably strong empirical evaluation

Weaknesses: - Main math section/derivation could be improved for clarity: it's a very simple idea but it is presented in an obfuscated way that requires a lot of energy from the reader to go through. - It feels Section 6 doesn't belong in this paper, and that the extra space could be used for extending the other parts of the paper. - The gains obtained by the proposed method modification are not that large.

Correctness: Claims, method and empirical methodology are correct.

Clarity: The paper is well written, except the main derivation is lacking clarity (see above).

Relation to Prior Work: Relation to prior work is clear.

Reproducibility: Yes

Additional Feedback: - Does the baseline in 5.3 also use 2 positive samples? - SimCLR with ResNet50 on ImageNet would have been a great addition to this paper and would make the results a lot stronger. (although it's not clear there's a big advantage over the baseline if the number of classes is 1000) - Add a disclaimer about the fact that the "positives" in the random negative samples and the "positives" in the correct term come from a different distribution: the former are views drawn from different images, while the latter are augmentations from the _same_ image.


Review 3

Summary and Contributions: The authors formulate a new objective for contrastive learning which addresses the issue of “false negative” examples (i.e. images from the same class which are treated as negative examples), while only sampling from the “positive” (i.e. images having the same label) and marginal distributions. Specifically, while standard contrastive objectives maximize the similarity of positive pairs normalized by the similarity to negative pairs sampled from the marginal, the authors propose a correction term to the denominator which accounts for the presence of such false negatives. Importantly, the correction term only requires sampling from the “positive distribution”, and doesn’t require explicit knowledge of the negative distribution. The authors validate their new estimator with a variety of representation learning experiments, in vision, NLP, and RL.

Strengths: The authors present a novel theoretical analysis which addresses the presence of “false negatives” in a set of negative examples sampled from the marginal (as is standard). This analysis results in an estimator which only requires samples from the positive distribution (i.e. images having the same label), and can therefore be approximated by sampling transformations of a single image. The resulting algorithm obtains competitive results on a variety of (albeit small-scale) benchmarks.

Weaknesses: The main weakness that I see with the paper is the mismatch between the theoretical analysis and the algorithm used in the experiments. The proposed estimator uses samples from the true “positive distribution” which consists of images from the same class. This is of course infeasible in a self-supervised setting where labels are unavailable. As a result, the authors approximate this distribution with the usual “positive distribution” which consists of random transformations of a single image. I understand that this two-step procedure is necessary to have an tractable analysis (using the true "positive distribution") and an experimental approach which is comparable to other self-supervised approaches (which use the approximate "positive distribution"), but the approximation of the “true positive” distribution by the other should be made explicit and discussed. There is only a fairly vague reference to this issue in Section 5.5, paragraph “Positive Distribution”, which does not acknowledge the mismatch between their theoretical analysis and their experimental setup. Regarding the experimental evaluation it appears for CIFAR-10 that the benefits of de-biasing the contrastive objective diminish with increasing batch size/number of negatives (Fig 4a). This effect is less pronounced or absent in STL-10 (Fig 4b). It would therefore be interesting to assess how these trends evolve by continuing to increase the number of negative samples as in MoCo. Since the use of a memory bank makes these large effective batch sizes readily available it seems important to assess the gains of the proposed method in this more realistic regime. Regarding the effect of adding more positive examples to obtain better estimates of the correction factor (Fig 4c) it seems as though an important baseline is missing. Given that these experiments require additional computation (M=2 requires three augmentations compared to the standard two), a very simple baseline would consist of simple running SimCLR for 50% more epochs. This additional computation is known to improve performance significantly (see Figure 9 of SimCLR paper) hence it would be useful to know whether the reported gains can simply be attributed to this additional computation. In summary, I encourage the authors to - properly explain and discuss the mismatch between their theoretical analysis and their empirical setup - Run the experiments in Figure 4a,b with larger numbers of negative examples. - Run SimCLR for longer training schedules, in proportion to the additional computation required by the additional positive samples, as an additional baseline. Alternatively, explain that these experiments are primarily a qualitative analysis and a proposal for a practical

Correctness: The empirical validation of the proposed estimator is sound.

Clarity: The paper is very well written.

Relation to Prior Work: A few references to prior art in contrastive representation learning are missing. Concurrently to MoCo, both PIRL and CPC v2 [A, B] found that self-supervised representation learning can surpass supervised representation learning, and should be acknowledged as such (e.g. in the sentence of introduction: "Recently, self-supervised representation learning algorithms that use a contrastive loss have outperformed even supervised learning [15, 27, 18, 2]"). SimCLR builds upon these works and furthers these gains. [A] Self-Supervised Learning of Pretext-Invariant Representations [B] Data-Efficient Image Recognition with Contrastive Predictive Coding

Reproducibility: Yes

Additional Feedback: POST REBUTTAL: The authors have provided a thorough rebuttal which addresses all of the concerns in my review. I have increased my score from 6 to 7. Congratulations on the great work!


Review 4

Summary and Contributions: The author proposed a debased contrastive learning algorithm to sample same-label data points. This is a general finding and empirically useful for the community to better understand and use CL algorithms. Unlike most works target on negative samples as a whole, this one points out more positive samples are needed. The author also shows the empirical results.

Strengths: This a well written theoretical paper. The proof path is clear. The idea of this paper is reasonable and potentially influenceable. The t-sne on M choice is a good empirical support of the paper idea.

Weaknesses: In the proof, the author assume N is a relatively large number (lemma 2), while in the experiments, N is not quite large and as N increases the difference between bias and best debias is minor. The pseudo code should be a M version, not a M=1 version, to give people more idea of implementation. The experimental improvement is really minor. The NLP experiments are not convincing as these dataset are quite small compared to CV experiments. It may be a good application filed for CL. A better choice may be large corpus, like yahoo question. I appreciate the author do the experiments on many different domain, but the NLP experiments is relatively rough.

Correctness: yes

Clarity: yes

Relation to Prior Work: yes

Reproducibility: Yes

Additional Feedback: I personally really like this paper. My major concern is the experimental results are insignificant. If the author can address my concerns, I would like to rise my score.

[Author Response · NeurIPS 2020]

We thank all reviewers for their helpful comments and feedback.

**Reviewer 1** 1. We will add the concurrent related works (CPCv2, MoCov2).

2. Parameter $\tau^+$ and $M$: We agree that picking $\tau^+$ could be domain-dependent and will add this clarification. Our
results show that larger $M$ consistently leads to better results regardless of domain. Scaling it up is indeed an interesting
future direction. We will add these points of discussion.

**Reviewer 2** 1. QT baseline: Yes, the baseline (QT) in 5.3 also uses 2 positive samples.

2. Full ImageNet: Indeed, with a larger number of classes, the number of collisions naturally shrinks. We will work on
adding larger-scale experiments.

3. Positive distribution: Section 5.5 provides a discussion about the "surrogate" positive distribution. We will state this
at the beginning of section 3 and section 5 to improve the clarity.

**Reviewer 3** 1. To improve the clarity, we will explicitly state the difference between true and approximated positive
distribution at the beginning of section 3 and section 5.

2. To probe the effect of debiased objective when the number of negative sample increases, we increase the batch size
to 512 ($N = 1022$) for the experiments in Figure 4ab and show the results in Figure 1 ab. The debiased objective still
outperforms the biased baseline when the negative sample size is doubled.

3. We run biased SimCLR for 50% more epochs (600 epochs) and compare it with the debiased objective ($M > 1$) in
Figure 4c. The results are shown in Figure 1 c. The debiased objective ($M > 1$) still outperforms the biased baseline
even it is trained with 50% more epochs.

4. We will add the missing references on contrastive learning, e.g., PIRL and CPC v2, in the related work section.

Figure 1: **Classification accuracy on CIFAR10 and STL10.** (a,b) Biased and Debiased ($M = 1$) SimCLR with larger
negative sample size $N$. (c) Comparison with biased SimCLR with 50% more training epochs (600 epochs) while
fixing the training epoch for Debiased ($M \geq 1$) SimCLR to 400 epochs.

**Reviewer 4** 1. Indeed, we derive the new objective by looking at the asymptotic version with large $N$. Interestingly,
and very useful, the objective also works with smaller $N$. The bound in Theorem 3 analyzes the approximation error
for smaller $N$.

2. Pseudocode: We show the $M = 1$ version to give a clear comparison with the standard (biased) contrastive loss. We
will add the M dependent version by changing the "pos" in line 8 of Figure 3 with an average of exponentials for $M$
positive samples.

3. We follow Kiros et al. [2015], Logeswaran and Lee [2018] in choosing these datasets. We agree that the improvements
in our NLP experiments are not as significant as the CV and RL experiments and we will work on extending our method
to other NLP benchmarks.

## References

Ryan Kiros, Yukun Zhu, Russ R Salakhutdinov, Richard Zemel, Raquel Urtasun, Antonio Torralba, and Sanja Fidler. Skip-thought
vectors. In *Advances in Neural Information Processing Systems*, pages 3294–3302, 2015.

Lajanugen Logeswaran and Honglak Lee. An efficient framework for learning sentence representations. *International Conference on
Learning Representations*, 2018.


[Meta-Review · NeurIPS 2020]

Reviewers unanimously agreed that this submission should be accepted -- it's well-written and proposes an interesting, well-motivated, and theoretically grounded idea. It's cheap and easy to implement and add to any existing contrastive learning approach, and the empirical evaluation showed strong results. Unsupervised contrastive learning is currently an area with plenty of interest among the NeurIPS audience so it is likely to be relevant/interesting to a large slice of the community. The authors' rebuttal addressed most of the reviewers' initial concerns. The paper could be further strengthened by evaluating the method in the full ImageNet-1K setting as this is the standard benchmark for most unsupervised visual representation learning work (even if the bias isn't large enough for the method to have much effect with 1K classes, this is useful for readers to know) -- the authors should strongly consider including these results in the camera-ready version of the paper.